# Role of Human Primary Renal Fibroblast in TGF-β1-Mediated Fibrosis-Mimicking Devices

**DOI:** 10.3390/ijms221910758

**Published:** 2021-10-05

**Authors:** Seong-Hye Hwang, Yun-Mi Lee, Yunyeong Choi, Hyung Eun Son, Ji Young Ryu, Ki Young Na, Ho Jun Chin, Noo Li Jeon, Sejoong Kim

**Affiliations:** 1Department of Internal Medicine, Seoul National University Bundang Hospital, Seongnam-si 13620, Korea; jasno1@daum.net (S.-H.H.); yunmi1202@hanmail.net (Y.-M.L.); R2759@snubh.org (Y.C.); she081792@gmail.com (H.E.S.); jyryu1022@gmail.com (J.Y.R.); kyna@snubh.org (K.Y.N.); mednep@snubh.org (H.J.C.); 2Department of Internal Medicine, Seoul National University College of Medicine, Seoul 03080, Korea; 3Program for Bioengineering, School of Engineering, Seoul National University, Seoul 08826, Korea

**Keywords:** renal fibroblast, fibrosis, TGF-β1

## Abstract

Renal fibrosis is a progressive chronic kidney disease that ultimately leads to end-stage renal failure. Despite several approaches to combat renal fibrosis, an experimental model to evaluate currently available drugs is not ideal. We developed fibrosis-mimicking models using three-dimensional (3D) co-culture devices designed with three separate layers of tubule interstitium, namely, epithelial, fibroblastic, and endothelial layers. We introduced human renal proximal tubular epithelial cells (HK-2), human umbilical-vein endothelial cells, and patient-derived renal fibroblasts, and evaluated the effects of transforming growth factor-β (TGF-β) and TGF-β inhibitor treatment on this renal fibrosis model. The expression of the fibrosis marker alpha smooth muscle actin upon TGF-β1 treatment was augmented in monolayer-cultured HK-2 cells in a 3D disease model. In the vascular compartment of renal fibrosis models, the density of vessels was increased and decreased in the TGF-β-treated group and TGF-β-inhibitor treatment group, respectively. Multiplex ELISA using supernatants in the TGF-β-stimulating 3D models showed that pro-inflammatory cytokine and growth factor levels including interleukin-1 beta, tumor necrosis factor alpha, basic fibroblast growth factor, and TGF-β1, TGF-β2, and TGF-β3 were increased, which mimicked the fibrotic microenvironments of human kidneys. This study may enable the construction of a human renal fibrosis-mimicking device model beyond traditional culture experiments.

## 1. Introduction

Renal fibrosis, the final common pathway of innumerable progressive kidney diseases, is characterized by an increase in extracellular matrix (ECM) protein production, a decline in matrix degradation, dysfunction of cell-matrix interaction, transformation of resident cells, and inflammatory cell infiltration. Transforming growth factor-β (TGF-β) is a multifunctional dimeric peptide that regulates biological processes such as cell proliferation, differentiation, and immunological responses [1]. Among the numerous fibrogenic factors, TGF-β plays a central role and has an anti-inflammatory effect. TGF-β1-deficient mice die of massive inflammation. Transgenic mice overexpressing TGF-β1 are virtually protected from developing renal fibrotic pathology, mainly through its anti-inflammatory activity [2]. The profibrotic and anti-inflammatory properties of TGF-β are double-edged swords regarding the therapeutic application of TGF-β inhibition. Remarkable efforts have been made to obstruct TGF-β action to hinder the progression of renal fibrosis [3]. Although many approaches have been utilized to combat renal fibrosis, an experimental model to evaluate currently available drugs is not ideal.

In general, animal studies play a pivotal role in preclinical tests in predicting pharmacokinetics and drug efficacy. However, various differences exist between animals and humans, which drives us to question the accuracy of animal-based drug efficacy and safety tests [4]. Previous reports have depended on preclinical animal studies during drug development to assess the nephrotoxicity and safety of drugs. The renal functioning of animals is more than twice that of the human kidneys. Hence, drugs are metabolized more quickly in animals than in humans, making it difficult to elucidate the results of drug toxicity tests using animal studies. Furthermore, the results of animal experiments cannot be used to predict drug responses in humans, as animals have different physiological functions. Animal-based renal fibrosis models have difficulties in reproducing human renal fibrosis; hence, research is underway to overcome current obstacles and optimize drug-discovery platforms [5].

Organ-on-a-chip (OoC) technology has emerged as a novel concept to solve these issues. This platform was designed considering the current shortcomings, and it has shown great promise in the field of nephrology [6]. In addition, OoC models hold great potential as a replacement for experimental animal models, provide a solution for inter-species differences, and can help end animal cruelty and ethical debates [7]. There are few in vitro models that simultaneously use primary renal fibroblasts with endothelial and epithelial cells, which are important cells in kidney fibrosis. The role of each cell in fibrosis is being studied separately; however, there is limited knowledge concerning the interaction of cells.

In the present study, we developed fibrosis-mimicking devices using human primary fibroblasts. We confirmed that TGF-β1 treatment has various effects on the renal fibrosis model. Notably, fibroblasts play a pivotal role as effector cells, promote the formation of a pro-fibrotic microenvironment, and secrete growth factors and cytokines. Therefore, based on this new method, fibroblasts may provide an ideal cell model system to study renal disease. We evaluated whether fibrotic renal disease-derived fibroblasts affect the integrity of cells cultured in three dimensions (3D). Finally, we present a proof-of-principle study that demonstrates the potential of human renal fibrosis-mimicking devices as a model for predicting the response to human renal fibrosis.

## 2. Results

### 2.1. Identification of Alpha Smooth Muscle Actin (α-SMA) and Keratin-8 (KRT-8) by TGF-β1 in HK-2 Cells

In the early phase, we examined known markers of fibrosis and the cytoskeleton in two-dimensional (2D) cultures. Immunofluorescent staining assays were performed to estimate α-SMA expression of a fibrosis and tubular epithelial to mesenchymal transition (EMT) marker and KRT8 levels as a cytoskeleton marker. Immunofluorescence staining showed that the expression of α-SMA was originally weak and KRT8 levels were high in the monolayer HK-2 epithelial cells. There was no significant alteration in fluorescence intensity by TGF-β1 (5 ng/mL) or the specific inhibitor treatment of HK-2 epithelial cells for 24 h (Figure 1).

### 2.2. KRT8 Expression Decreased by TGF-β1 on a Chip

Next, we evaluated the expression levels of α-SMA and KRT8 in 3D-cultured HK-2 and the total length of GFP-human umbilical-vein endothelial cells (HUVECs) treated with TGF-β1 and a specific inhibitor. To accomplish this, we first established a tissue chip pattern comprising two cell types, HK-2 epithelial cells and GFP-HUVECs. Following confirmation of the construct, we exposed the tissue chip with TGF-β1 (5 ng/mL) or a specific inhibitor (10 µm SB431542) for 24 h.

As shown in Figure 2A–C, the expression of α-SMA was not specifically changed by TGF-β1 or the specific inhibitor treatment. However, the expression of KRT8 was reduced in the two-cell type co-cultured chip. Moreover, the addition of a TGF-β1 inhibitor increased the expression of KRT8.

We examined the formation of a 3D tubular capillary-like network by TGF-β1 treatment in HUVECs, as endothelial cells can spontaneously form a 3D tubular capillary-like network. In accordance with our culture conditions, the formation of thick lines was significantly decreased. However, capillary-like thin lines were increased in the TGF-β1 treated GFP-HUVECs. In the case of the TGF-β1 inhibitor treatment on the 3D-cultured chip, the total length of the thick and thick lines showed a reverse tendency to TGF-β1 treatment. An increase in the length of the thick line was observed compared to the untreated control group. A similar pattern was observed in the diameter of the thick line in GFP-HUVEC after TGF-β1 treatment and inhibitor treatment. The diameter of the GFP-HUVEC was suppressed by the TGF-β1 treatment but increased irrespective of the thickness of the line by the inhibitor compared to both the controls and TGF-β1. The total length and diameter of the thick line were dramatically increased by TGF-β1 inhibitor treatment compared to the untreated control and TGF-β1 treatment. Notably, the obtained results indicated an increased density of tiny vessels in the TGF-β-treated group compared to their untreated counterparts. In contrast, the density of thick vessels in the TGF-β1 treatment group was lower than that in the untreated control group (Figure 2D,E).

### 2.3. TGF-β1 Affects Three Cell Types of the Chip with a Multi-Compartment Structure

To study the TGF-β1 responses of the proposed 3D chip, we performed immunostaining on each compartment separately (Figure 3). Primary human renal fibroblasts were utilized in this analysis. The samples were analyzed using FACS and confirmed to be positive for the fibroblast marker antibody. Following incubation of the three cell types with TGF-β1 for 24 h, immunostaining was performed to assess the expression of α-SMA and KRT8. The expression of α-SMA was upregulated by TGF-β1 treatment, and opposite results were observed with TGF-β1 inhibitor treatment. Conversely, the expression of KRT8 was downregulated by TGF-β1 and showed an opposite response to the TGF-β inhibitor (Figure 4A–C).

As depicted in Figure 4D–G, the length of the capillary-like thin line was significantly increased by TGF-β1 treatment in the GFP-HUVECs. However, the thickness of the thick line was decreased in the TGF-β1 treated HUVECs. The diameter was reduced on average for the thin and thick vessels in the HUVECs. In the case of the TGF-β1 inhibitor treatment on the 3D-cultured chip, the total length of the thick and thick lines showed an opposite trend to TGF-β1 treatment. Further, the length of the thick line was increased compared to the untreated control group.

### 2.4. TGF-β1 Modulates Inflammatory Mediator and Growth Factors

Next, we investigated the inflammatory cytokines and growth factors in the supernatant; IL-1β, FGF-2, TGF-β2, and TGF-β3 protein secretion was dramatically higher in 3D-cultured chips than the 2D fibroblast cultured well, even though they had the same total cell count. However, TGF-β1 levels were similar, as the 2D- or 3D-cultured models were treated with similar concentrations of TGF-β1 (Figure 5A–E). Particularly, the release of FGF-2 from human primary renal fibroblasts was 20.9 ± 0.4 pg/day/5 × 10^5^ cells of the 2D mono-layered culture and 3094.8 ± 0.2 pg/day/5 × 10^5^ cells including the HUVECs and HK-2 of the 3D-cultured chip in the untreated control. The secretion of FGF-2 by TGF-β1 stimulation was 31.0 ± 0.2 pg/day/5 × 10^5^ cells for the 2D culture and 3683.9 ± 0.8 pg/day/5 × 10^5^ cells for the 3D-cultured chip. This increase in FGF-2 protein production in the 2D monolayer culture was paralleled by an increase in the 3D-cultured chip model by TGF-β1 treatment (Figure 5B).

Furthermore, we estimated the mRNA expression of pro-inflammatory cytokines such as IL-1β, IL-6, IL-8, and TNF-α. The results were significantly downregulated by recombinant human TGF-β1 treatment in the 3D-cultured chip as an anti-inflammatory property of TGF-β. We found that mRNA expression was upregulated by the TGF-β1 inhibitor (SB431542). Furthermore, inhibitor-mediated expression of IL-6, IL-8, and TNF-α was specifically downregulated compared to in the untreated control (Figure 6A–D). The mRNA level of VEGF, a fibrotic factor, was significantly increased by TGF-β1 treatment. The mRNA level of VEGF was decreased by the TGF-β1 inhibitor (Figure 6E), whereas the mRNA expression of IL-10 with anti-fibrotic effect was decreased in fibrotic conditions induced by TGF-β1. In contrast, IL-10 mRNA expression was increased by TGF-β1 inhibitor treatment (Figure 6F).

## 3. Discussion

TGF-β1 plays an important role in matrix accumulation in renal fibrogenesis and inhibits proliferation in most cells, including glomerular epithelial and endothelial cells, as well as tubular epithelial cells [1]. However, TGF-β1 induces proliferation in human renal fibroblasts via induction of basic fibroblast growth factor (FGF-2) [8].

In the present study, we established a fibrosis-mimicking device using three essential cellular components, including human primary renal fibroblasts, renal tubular epithelial cells, and human endothelial cells, and evaluated it as a renal fibrosis model based on TGF-β1 stimulation. To generate renal fibrosis-on-a-chip, we established a novel 3D-culture system designed with three separate parts of tissue, aiming to overcome obstacles associated with renal disease modeling. Our study demonstrated that TGF-β1 induced cellular responses, including tubular epithelial-mesenchymal transition (EMT) of epithelial cells, microvascular alteration of endothelial cells, and cytokine changes in renal fibroblasts. This is the first study to introduce a fibrosis-mimicking device comprising primary fibroblasts derived from human kidney and stimulated fibroblasts isolated from patients with TGF-β1, a known inducer of fibrotic responses generally used as the standard to mimic fibrosis in cell cultures [9]. These results suggest that TGF-β1-treated epithelial cells are converted to a mesenchymal cell-like phenotype. When TGF-β1 is administered to fibroblasts, various cytokines are altered to reflect the induction of fibrotic processes. It is apparent that in the renal fibrosis model, fibroblasts are the main force behind the development of the established culture system through TGF stimulation.

Davis et al. reported that the 3D culture of endothelial cells is sufficient for the cells to invade and form a lumen and tube network, following an intact organism in vivo [10,11]. Several groups have reported that 3D-culture models exhibit higher levels of nephrotoxicity than 2D cell-culture models [12]. Furthermore, the expression of nephrotoxicity- and inflammation-related genes was significantly higher in the renal 3D cell-culture model than in a typical 2D culture [13].

In this study, the culture media of human renal fibrosis-on-a-chip had a significantly higher level of cytokines compared to that in 2D culture. Our group has shown that patient-derived fibroblasts can recreate native renal tissues, including endothelial and epithelial cell layers, through our suggested fibrosis-on-a-chip model. Consequently, these findings indicate that the renal fibrosis-on-a-chip model accurately simulates the microphysical environment of the human kidney. In addition, because the fibroblasts cultured in the chip are derived from patients, other clinically relevant patient models can be easily and clearly obtained in a safe manner. Therefore, the human renal fibrosis-on-a-chip model system can allow for the development of personalized medicine with sensitive measurements of epithelial and endothelial functions. In our experiments, fibrosis and angiogenesis were decreased in the TGF-β inhibitor-treated groups. TGF-β inhibitors are natural antagonists of TGF-β, which have been demonstrated using various renal disease models. It appears that it is possible to conduct experiments to confirm the effectiveness of treatment in disease models using therapeutic agents such as this TGF-β inhibitor. Therefore, this chip has provided a means to develop an improved model of renal fibrosis to measure cell morphology and function and might be useful in performing nephrotoxicity assessments for pharmaceuticals.

TGF-β is the predominant profibrotic factor in various kidney diseases [14] and plays a role in angiogenesis [15]. The angiogenic effects of TGF-β are very complex and can have either proangiogenic or antiangiogenic activity depending on the setting and various regulatory factors [16]. Angiogenic growth factors such as VEGF and FGF-2 induce angiogenesis in vivo and appear to play a central role in the modulation of angiogenesis in vivo [17,18,19,20]. Some evidence indicates that these two factors can act synergistically to promote endothelial cell morphogenetic events [21,22]. We have provided evidence that VEGF mRNA expression and FGF-2 protein secretion are significantly elevated by TGF-β1 in 3D chips compared to 2D culture.

The limitation of our fibrosis models might be that they were created in a relatively short time period, 24 h. TGF-beta have pro-fibrotic and anti-inflammatory properties within an early fibrosis process, not in the chronic process. Conventionally, TGF-β has been primarily utilized in cytokine-induced fibrosis models. However, since only one cytokine cannot be represented in vivo, fibrosis induction models using several additional candidate substances such as TGF-β and BMP7 have been proposed. However, two or three cytokines are still not sufficient to reproduce in vivo. We used TGF-β as the starting point; however, we tried to implement an in vivo microenvironment on the chip by inducing a more diverse cytokine storm. Fibroblast, a major cellular component of fibrosis, was introduced, and secondary stimulation of more diverse cytokines (IL-1β, TNF-α, b-FGF, TGF-β1, TGF-β2, and TGF-β3) was induced in fibroblasts stimulated with TGF-β. Thus, an environment similar to fibrosis in the human body was reproduced.

## 4. Materials and Methods

### 4.1. Cell Cultures

Kidney fibroblasts (KFs) were isolated from biopsies of the normal tissue portion of renal cell carcinoma patients with an estimated glomerular filtration rate (eGFR) > 60 mL/minute/1.73 m^2^ after patients had given their consent to a second biopsy for research purposes. KFs were cultured in fibroblast growth medium (FGM-2, Lonza, Switzerland), and passages 3 to 4 were used for the experiments. Renal fibroblasts were scraped from the culture for 1–2 weeks until the cells formed a monolayer. To eliminate epithelial cells contaminating the culture, fibroblasts were selected using magnetic anti-fibroblast beads (Miltenyi Biotec Inc., Auburn, CA, USA) after the first passage. Cells were detached using 0.05% trypsin-ethylenediaminetetraacetic acid (EDTA) (Welgene, Gyeongsan, Korea), incubated with anti-fibroblast beads, and separated using a magnetic column, and the flow-through was collected. The purified primary fibroblasts were characterized by fluorescence-activated cell-sorting (FACS) analysis using PE-conjugated anti-fibroblast antibodies. The gentle magnetic-activated cell-sorting (MACS) Dissociator was used for gentle kidney tissue mincing. An Octo MACS™ Magnetic Separator with MACS LS columns (with plungers) applied for purification of microbead-labeled cells were purchased from Miltenyi Biotec Inc. (Auburn, CA, USA). Gentle MACS C tubes (Miltenyi Biotec Inc., Auburn, CA, USA) were used for kidney mincing for cell separation. MACS Smart Strainers (70 μm) were obtained from Miltenyi Biotec Inc. (Auburn, CA, USA). Column buffer was used to wash columns and resuspend cell pellets. The buffer was prepared as a solution containing 0.5 % bovine serum albumin (BSA) with 2 mM EDTA in phosphate buffered saline (PBS) (pH 7.4) and purchased from Miltenyi Biotec Inc. (Auburn, CA, USA). Green fluorescent protein-expressing human umbilical-vein endothelial cells (GFP-HUVECs, Lonza, Switzerland) were cultured in Endothelial Growth Medium (EGM-2, Lonza, Switzerland), and cells in passages 3 to 4 were used. Human proximal tubular cell line (human kidney-2 (HK-2) cells were obtained from the American Type Culture Collection (ATCC). The cell line was grown in high-glucose Dulbecco’s modified Eagle’s medium (DMEM) supplemented with 10% fetal bovine serum (FBS), penicillin (100 U/mL), and streptomycin (100 µg/mL). Reagents were purchased from Gibco (Rockville, MD, USA). All cells were detached using 0.05% Trypsin-EDTA and maintained in a humidified incubator at 37 °C and 5% CO_2_. KFs were re-suspended in FGM-2 at a cell concentration of 4 × 10^6^ cells/mL. Human kidney tubular epithelial cells (HK-2) were re-suspended at a concentration of 2 × 10^6^ cells/mL in high-glucose DMEM. GFP-HUVECs were re-suspended at a concentration of 2 × 10^6^ cells/mL in EGM-2.

### 4.2. Reagents

The matrix was composed of fibrin gels, including commercially available fibrinogen from bovine plasma, aprotinin from bovine lung, and thrombin from bovine plasma. The aforementioned three materials were purchased from Sigma (St. Louis, MO, USA). Fibrinogen was dissolved in 1x PBS in a water bath (37 °C) for 30 min at a concentration of 10 mg/mL. Thrombin and aprotinin were dissolved in deionized water at a concentration of 50 units/mL and 4 TIU/mL. The three solutions were sterilized by filtering through a 0.22 μm filter. Recombinant human TGF-β1 was obtained from PeproTech EC Ltd. (London, UK). A specific inhibitor of TGF-beta receptor kinase, SB431542, was purchased from Tocris Cookson, Inc. (Ellisville, MO, USA).

### 4.3. Cell Patterning in the Device

Before cell seeding to facilitate tight bonding, the device was treated for 1 min in a plasma machine (Femto Science Inc., Suwon, Korea) with a power of 70 W, 50 Hz. Plasma treatment induced surface hydrophilicity and helped facilitate gel patterning and media loading. To avoid changes in hydrophilicity, experiments were performed within 30 min after plasma treatment. Different cellular patterning within the devices can be done in a highly customizable fashion. It was important to determine the composition and concentration of the different cell types to be patterned beforehand. For cellular hydrogel suspensions, 37.5 µL of cell suspension and a separated aliquot of 12.5 µL of 10 mg/mL fibrinogen solution (250 µL of 10 mg/mL fibrinogen solution with 40 µL of 4 TIU/mL aprotinin) were prepared to be mixed immediately prior to loading. Immediately prior to loading, the 37.5 µL of cell suspension was mixed with 12.5 µL of 10 mg/mL fibrinogen until homogenized. The 50uL of hydrogel was mixed with a 1 uL droplet of thrombin (0.5 unit/mL). Immediately after mixing, thrombin with a fibrin suspension of GFP-HUVECs (cell concentration of 2 × 10^7^ cells/mL) was injected into the outer edge of the outer channel (Figure 3, Gel A). We waited 3 min for fibrinogen crosslinking to be complete. Immediately after mixing, thrombin with a fibrin suspension of KFs (cell concentration of 4 × 10^6^ cells/mL) was placed on each side of the reservoir floor per well (Figure 3, Gel C). This fibrin suspension of KFs was allowed to crosslink at room temperature for 3 min. A cell suspension of 10 µL human renal proximal tubular epithelial cells (HK-2) was injected the injection port of the inner channel (Figure 3, Gel B). To attach HK-2 to the wall of the fibrin gel in the GFP-HUVECs channel, the device was turned 90 degrees and placed in the 5% CO_2_ incubator for 30 min at 37 °C. The HK-2 settled down and formed a cell sheet on the side of the fibrin gel.

After attaching the HK-2, the device was filled with EGM-2. The device was incubated for 3 days at 37 °C in a 5% CO_2_ incubator. The medium was changed with or without 5 ng/mL TGF-β1 and with 5 ng/mL TGF-β1 and 10 µm TGF-β1 inhibitor after three days of incubation (Figure 3).

### 4.4. Immunocytofluorescence

Co-cultured tissues in the device were fixed with 4% (*w*/*v*) paraformaldehyde solution (Biosesang, Seongnam, Korea) in PBS (Welgene, Gyeongsan, Korea) for 20 min, followed by permeabilization with a 20 min immersion in 0.15% Triton X-100 (Sigma, St. Louis, MO, USA). The samples were then treated with 3% BSA (Sigma, USA) for 1 h. The cells were incubated with antibodies purchased from Abcam. The primary antibodies were anti-cytokeratin 8 and anti-α-SMA, left for 2 days at room temperature (RT). Alexa Fluor 647 conjugated donkey anti-rabbit IgG (H + L) was used as the secondary antibody. DNA labeling was performed with a 1:250 dilution of Hoechst 33342 (Thermo Scientific, Rockford, IL, USA) for 3 h at RT. Images were collected using a Zeiss LSM 710 confocal laser microscope. The mean fluorescence intensity (MFI) was measured using ImageJ and a fluorescence-intensity analyzer.

### 4.5. Multiplex Analysis of Cytokines

The MSD V-Plex Cytokine and Angiogenesis Panel 1 Human kit (Rockville, MD, USA) was used to measure interleukin-1 beta (IL-1β) and basic fibroblast growth factor (b-FGF) concentrations in single samples, according to the manufacturer’s instructions. All samples were assessed for total protein levels of IL-1β and b-FGF. The amount (in pg/mL) was quantified from the standard curve of each measurement.

### 4.6. Multiplex Bead Immunoassay

Cytokine concentrations were analyzed using a multiplex bead immunoassay system (Procarta Cytokine Assay Kit; Affymetrix, Inc., Santa Clara, CA, USA), based on multiplexing technology (xMAP; Luminex, Austin, TX, USA), according to the manufacturer’s instructions. The data were acquired using a Luminex-compatible workstation and its manager software (Bio-Plex workstation and version 6.0 software; Bio-Rad, Tokyo, Japan), according to the manufacturer’s instructions. Human TGF-β1, TGF-β2, and TGF-β3 related to the fibrosis process were analyzed simultaneously in the diluted samples. The isoforms of TGF-β were detected using Bio-Plex 200 Systems and commercially available Bio-Plex Pro™ Human TGF-β assays (Bio-Rad Laboratories, Inc); based on the information provided by the manufacturer. The multiplex assay kit can quantitatively measure multiple cytokines from the supernatant with a lower limit of detection of 1 pg/mL per cytokine. Each sample was run as a single measurement for a limited quantity of the collected supernatant.

### 4.7. Real-Time PCR

RNA extraction was performed using TRIzol Reagent (Invitrogen Life Technologies, Carlsbad, CA, USA). Master Mix and a Revert Aid First Strand cDNA Synthesis Kit was used for the reverse transcription of the total RNA. qPCR was performed using the Applied Biosystems™ PowerUp™ SYBR™ Green Master Mix (Thermo Fisher Scientific, Inc.). The specific primers were prepared by Bioneer (Daejeon, Korea) and used in this experiment as follows. The primer sequences are summarized in Table 1.

Ten microliters of PowerUp SYBR Green Master Mix, 4 µL of cDNA, and 2 pmol of each primer were used for real-time PCR in a final volume of 20 µL. The reaction was carried out at 95 °C for 1 s and 60 °C for 20 s for 45 cycles after denaturation at 95 °C for 20 s. PCR was performed in duplicate or triplicate for each sample. cDNA levels were determined using a standard curve of cycle thresholds. All data for each cDNA were within the corresponding standard curve. The data obtained were normalized to β-actin cDNA.

### 4.8. Statistical Analysis

Quantitative data are presented as mean ±SD or SE of at least three independent experiments. Statistical analyses were performed using a two-tailed Student’s t-test. We considered a significant difference only at a 95% confidence level or higher (*p* < 0.05).

## 5. Conclusions

In summary, the protocols in this study enabled the construction of human renal fibrosis-mimicking devices as a model and showed various effects that are not possible in 2D-culture experiments. We believe that the evaluation strategy proposed in this study will lead to a better understanding of the detailed fibrosis mechanisms involved in pathological changes during renal disease. The developed 3D renal fibrosis-on-a-chip could be used as a potential in vitro model, which will bring us closer to creating an enhanced kidney-on-a-chip model.

## Figures and Tables

**Figure 1 ijms-22-10758-f001:**
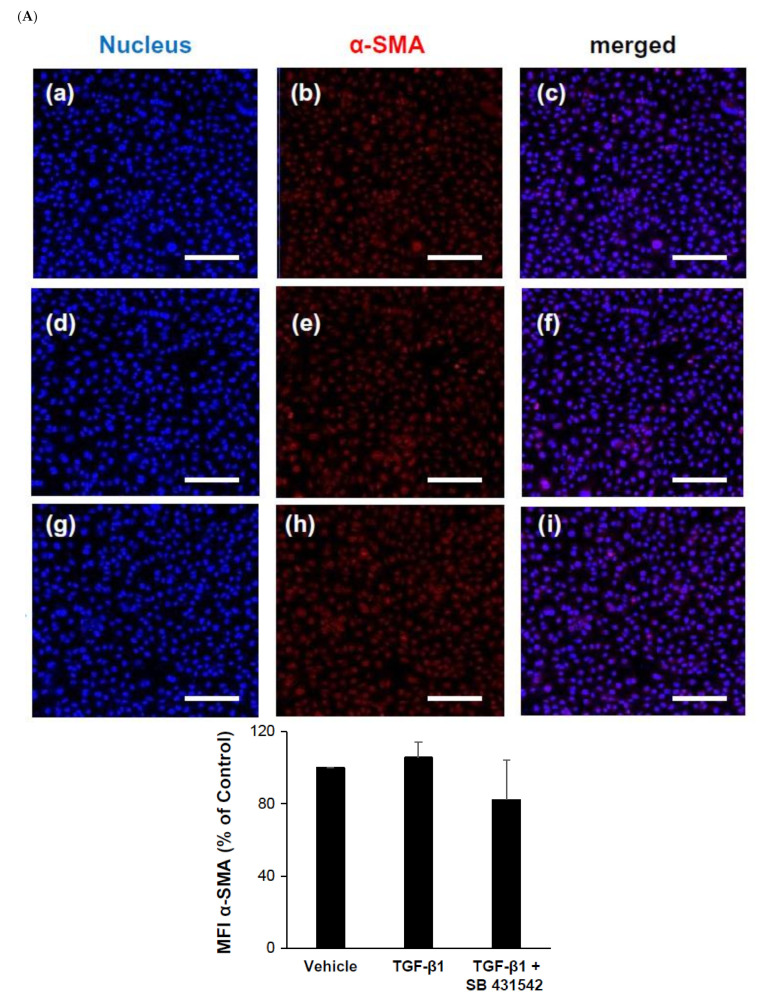
Immunofluorescence shows that the expression of alpha-SMA and keratin-8 (KRT8) were maintained by treatment of TGF-β1 or the inhibitor in HK-2 cells. (**A**) The expression of alpha-SMA and (**B**) CCK-8 had no effect on HK-2 cells by TGF-β1 (5 ng/mL) or the inhibitor. Cells were originally plated at a density of 1 × 10^5^ per well (**a–c**) untreated control and (**d–f**) stimulated with 5 ng/mL TGF-β1 or (**g–i**) 10 µM inhibitor (SB 431542) for 24 h and fixated with 4% paraformaldehyde. The cells were then stained with anti-α-SMA and anti-cytokeratin-8 for 20 min. Co-staining with Hoechst dye H33342 to identify cell nuclei was performed. Scale bars in micrographs indicate 200 µm.

**Figure 2 ijms-22-10758-f002:**
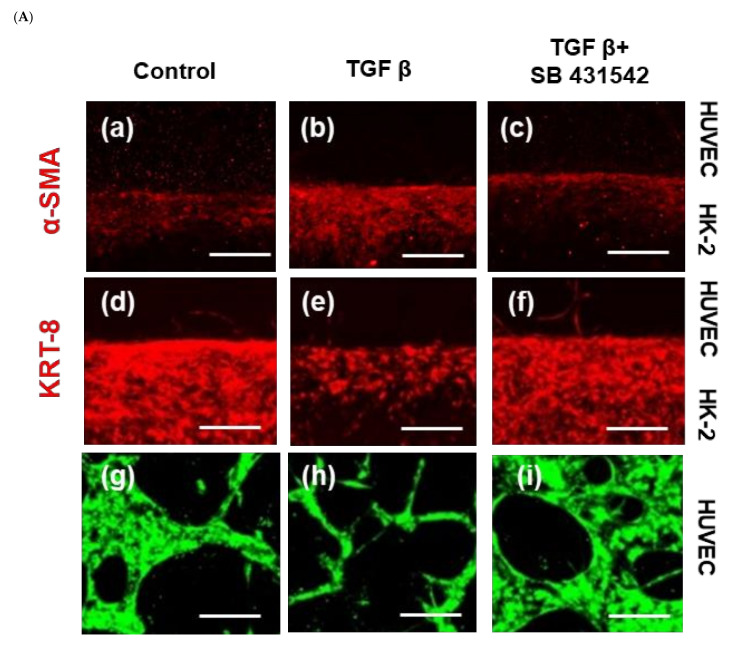
KRT8 expression in HK-2 and total length and diameter of the HUVECs. **(A)** Immunofluorescence shows that the expression of α-SMA was maintained and KRT8 expression was dramatically decreased by treatment of TGF-β1 in HK-2. Cells were plated and stimulated with 5 ng/mL TGF-β1 or 10 μm inhibitor (SB 431542) for 24 h and fixated with 4% paraformaldehyde. The cells were stained with anti-α-SMA and anti-cytokeratin-8 for 20 min. Co-staining with Hoechst dye H33342 to identify cell nuclei was performed. The expression of α-SMA (**a–c**) had no alteration but KRT8 (**d–f**) in HK-2 cells and GFP in HUVEC (**g–i**) expression were significantly changed by TGF-β1 (5 ng/mL) and the inhibitor (**B**,**C**). In the total length of the HUVECs, the thin vessel was increased but the thick vessel was decreased by TGF-β1. The diameter was increased in both the thin and thick vessels. These results were reversed by the inhibitor SB431542 (**D**,**E**). Scale bars in the micrographs indicate 100 µm. * *p <* 0.05, ** *p <* 0.01, *** *p* < 0.001 versus the control group; ## *p* < 0.01, ### *p* < 0.001 versus the TGF-β1 group. Each value represents three technical replicates of each of the three biological replicates. Statistical significance of the length compared to the non-treated cells is represented in the graph. Thin vessels mean a length shorter than 50 µm and thick vessels represent a length longer than 50 µm.

**Figure 3 ijms-22-10758-f003:**
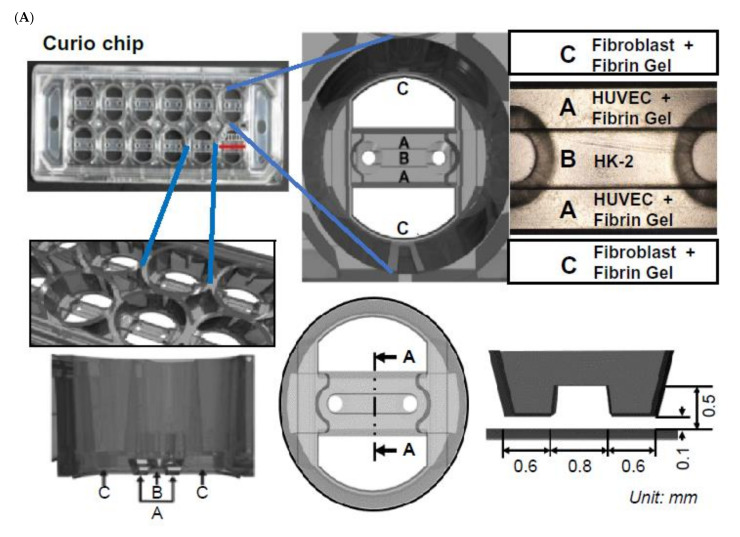
Sample image with device schematic demonstration of the fabrication and experimental schedule. (**A**) The top two images show the layout. The left image shows an overall top-down photo of the device. The bottom images show the cross-section. Schematic view detailing the dimensions of the liquid guides. (**B**) This figure describes the experimental schedules for fibrosis-mimicking devices. Four-step loading process for each well. Location of each hydrogel patterning area, as well as the placement of the media in top-down and isometric cross-section. (1) A total of 1.5 µL of hydrogel 1 is spontaneously guided into the central channel. (2) Central channel is filled with 5 µL of hydrogel. (3) A total of 10 µL of hydrogel 3 are patterned on the reservoir floor. (4) A total of 200 µL of media are dispensed. Red scale bar = 9 mm.

**Figure 4 ijms-22-10758-f004:**
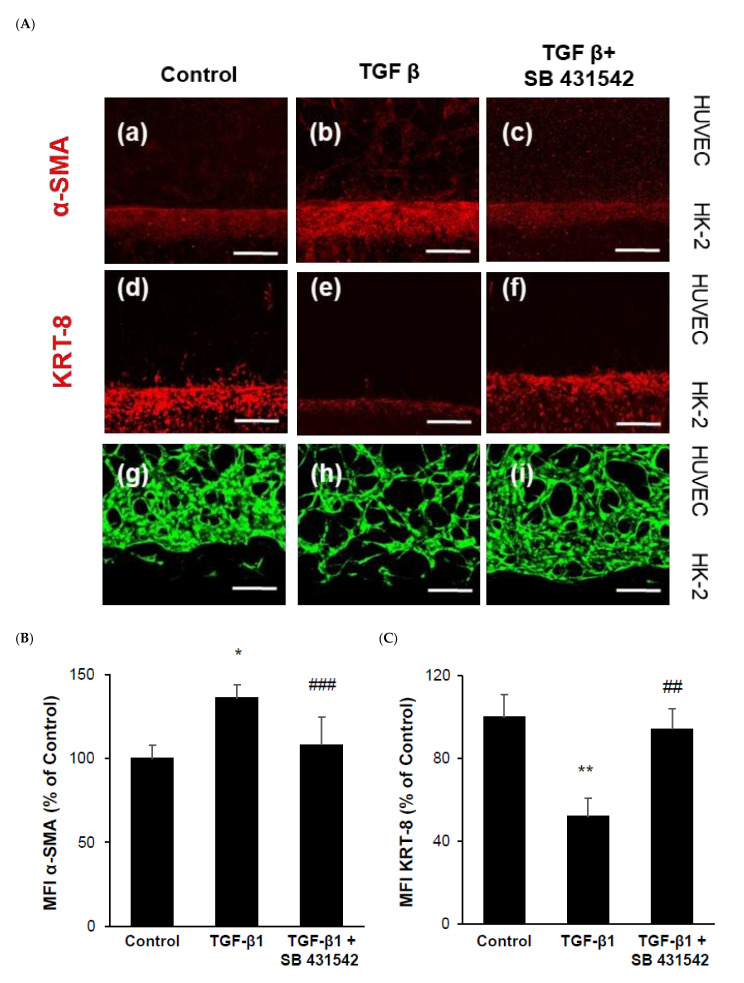
Alteration of 3D-cultured HK-2 and HUVECs with primary renal fibroblasts. Total cells were plated at a density of 5 × 10^5^ per 3D chip and stimulated with 5 ng/mL TGF-β1 or 10 μm inhibitor (SB 431542) for 24 h and fixated with 4% paraformaldehyde. (**A**) Cells in the 3D chip were stained with anti-α-SMA and anti-cytokeratin-8. The expression of α-SMA (**a**–**c**) was increased and KRT8 (**d**–**f**) expression was decreased by TGF-β1 (5 ng/mL) significantly. The total length of the HUVECs: thin vessels were dramatically increased and thick vessels were decreased by TGF-β1 (**D**,**E**). The diameter was decreased for thin and thick vessels (**F**,**G**). These results were reversed by the inhibitor SB431542. Scale bars in micrographs indicate 100 µm. * *p <* 0.05, ** *p* < 0.01, and *** *p* < 0.001 versus the control group; # *p* < 0.05, ## *p* < 0.01, and ### *p* < 0.001 versus the TGF-β1 group. Each value represents three technical replicates of each of the three biological replicates. Thin vessels mean a length shorter than 50 µm and thick vessels represent a length longer than 50 µm.

**Figure 5 ijms-22-10758-f005:**
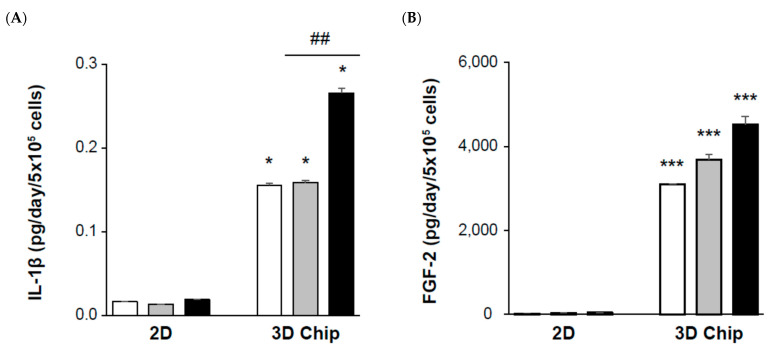
Comparison of 2D- and 3D-cultured models as renal fibrosis-mimicking platforms. Cells were plated at a density of 5 × 10^5^ per 2D well or 3D chip and stimulated with 5 ng/mL TGF-β1 or 10 µm inhibitor (SB 431542) for 24 h. Primary human renal fibroblasts were used in the 2D-cultured and in the 3D-cultured model. Fibroblasts were utilized at a density of 8 × 10^4^ cultured with HUVECs and HK-2. The primary human renal fibroblast-to-HUVEC ratio, or F:H ratio, was 1:5 for assessing fibrosis. HK-2 cells were used at a density of 2 × 10^4^. Detection of (**A**) IL-1β, (**B**) basic fibroblast growth factor (FGF-2), and (**C**–**E**) TGF-beta 1, 2, and 3 in the supernatant was performed by multiplex analysis of cytokines and multiplex bead immunoassay. Each bar represents the mean ± SE. * *p <* 0.05, *** *p* < 0.001 versus 2D; ## *p* < 0.01, ### *p* < 0.001 versus TGF-β1.

**Figure 6 ijms-22-10758-f006:**
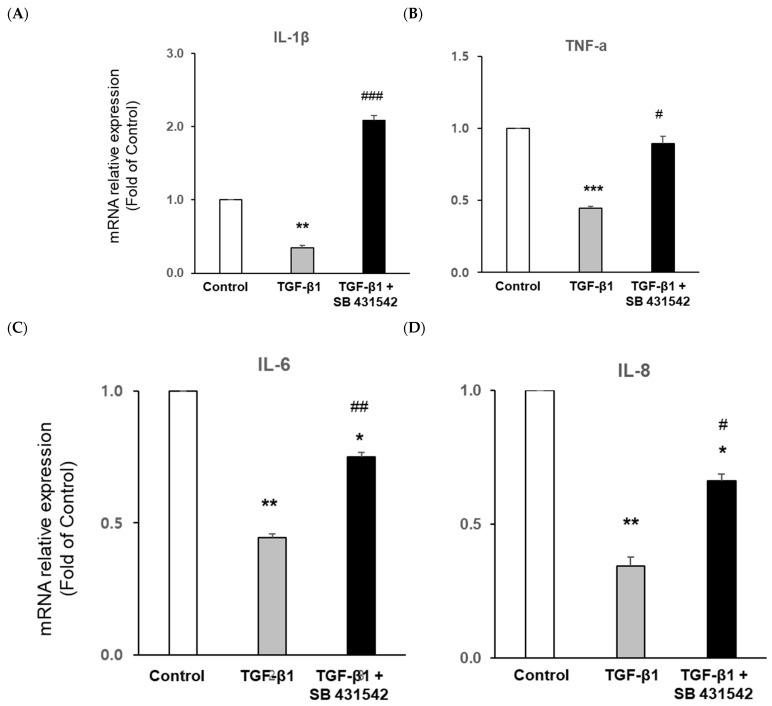
Real-time PCR shows the overall mRNA expression of the 3D chip. Cells were plated at a density of 5 × 10^5^ per 3D chip and stimulated with 5 ng/mL TGF-β1 or 10 μm inhibitor (SB 431542) for 24 h. Total RNA extracted from the 3D-cultured chip. The mRNA expression of (**A**) IL-1 β, (**B**) TNF-α, (**C**) IL-6, and (**D**) IL-8 represents the anti-inflammatory effect of TGF-β1 treatment. (**E**) VEGF expression as an angiogenic factor was increased. (**F**) IL-10 expression as an anti-fibrotic factor was significantly decreased by TGF-β1. * *p <* 0.05, ** *p <* 0.01, *** *p* < 0.001 versus control; # *p* < 0.05, ## *p* < 0.01, ### *p* < 0.001 versus TGF-β1.

**Table 1 ijms-22-10758-t001:** Primer sequences used for quantitative real-time PCR.

Gene	Forward Sequence (5′-3′)	Reverse Sequence (5′-3′)
IL-1 β	ATGATGGCTTATTACAGTGGC	GTCGGAGAGATTCGTAGCTGGA
IL-6	AAAGAGGCACTGGCAGAAA	TTTCACCAGGCAAGTCTCCT
IL-8	GACCACACTGCGCCAACAC	CTTCTCCACAACCCTCTGCAC
TNF-α	GAGTGACAAGCCTGTAGCCCA	GCAATGATCCCAAAGTAGACC
VEGF	TGCAGATTATGCGGATCAAACC	TGCATTCACATTTGTTGTGCTGTAG
IL-10	TTGCCAAGCCTTGTCTGA	AGGGAGTTCACATGCGCCT
β -actin	TCACCATTGGCAATGAGCGG	AGTTTCGTGGATGCCACAG

## Data Availability

The data presented in this study are available on request from the corresponding author. The data are not publicly available due to [Patent issue].

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
