# Peer review of "Role of Human Primary Renal Fibroblast in TGF-β1-Mediated Fibrosis-Mimicking Devices"

_ijms, 2021, doi:10.3390/ijms221910758_

Round 1

Reviewer 1 Report

In this manuscript, the authors developed fibrosis-mimicking models using 3D co-culture devices which designed with three separate epithelial, fibroblastic, and endothelial layers. They evaluated the effects of TGF-β and TGF-β inhibitor treatment on this renal fibrosis model and confirmed that TGF-β1 treatment has various effects on this model. The topic has merit because their results demonstrate the potential of human renal fibrosis-mimicking devices as a model for predicting the response to human renal fibrosis.

However, the manuscript is not yet suitable to be publish because part of the study is not enough well-conducted, and there are some problems need to address. These issues are discussed below.

Comments follow:

  1. First, a lot of the quantitative measurements of protein expressions in this manuscript by the authors were using detection of the fluorescence intensity in the immunofluorescence staining assay. However, the estimation of MFI is a relative semi-quantitative test which is not accurate enough as the only verification method for protein expressions. In addition, the authors also did not specify the number of samples used, the area selected for each sample, and the measurement conditions. This part requires more precise measurement methods such as WB or qPCR to do further proof.

  1. Second, the author did not explain how to define the standard of thin and thick vessels, since the formation of a 3D tubular capillary-like network is an important experimental result of the author, they should explain in more detail. In addition, the author should examine the effects of SB4315422 on their experiments separately. Because the effects of TGF-β1+TGF-β1 inhibitor group is much higher than that of the control group, so the effects of SB4315422 alone

  1. The authors described in conclusion that their 3D renal fibrosis-on-a-chip is better than the 2D culture experiments, but did not clearly explain what the characteristics of the 3D model are and how their model to simulate the situation of real kidney tissue? What the different and advantage is as comparing with co-culture together of 3 types of cells? Dose it belong 2D or 3D model in the 2.2 experiments on a chip? How is this different from the 2.3 experiments? just the addition of fibroblast in chip? This requires detailed explanation.

  1. The authors need to explain which cell line they used for isolating mRNA in the mRNA expression test, HK2 or HUVECs? In addition, they also need to explain how the test determines the diameter of the thick and thin vessels in HUVECs. Is the increased number of cells taken into consideration?

  1. Although the author tested the HUVEC cell growth and cytokine growth factor expressions of TGF treatment in their 3D model, they did not clearly explain the exact effects of TGF-β1 on kidney fibrosis. By their results, is the fibroblast enhanced the TGF-β1 effects on renal fibrosis or TGF-β1 reduce the renal fibrosis in 3D model? The author seems to have only given an ambiguous description here, and make their conclusion about “3D renal fibrosis-on-a-chip could be used as potential in vitro model" become weak.

  1. The immunofluorescent figures of the author showed are not clear enough, and higher resolution pictures should be provided for identification.

Author Response

Response to Reviewer 1 Comments

In this manuscript, the authors developed fibrosis-mimicking models using 3D co-culture devices which designed with three separate epithelial, fibroblastic, and endothelial layers. They evaluated the effects of TGF-β and TGF-β inhibitor treatment on this renal fibrosis model and confirmed that TGF-β1 treatment has various effects on this model. The topic has merit because their results demonstrate the potential of human renal fibrosis-mimicking devices as a model for predicting the response to human renal fibrosis.

We are grateful for this valuable comment and appreciate the positive feedback. Several statements that we made were more ambiguous than intended and we have adjusted the text to be clearer. We realize that the initial text may have been unclear and acknowledge that more detail is needed, we have added statements specifically and revised especially methods description in 4.3.

However, the manuscript is not yet suitable to be publish because part of the study is not enough well-conducted, and there are some problems need to address. These issues are discussed below.

Comments follow:

  1. First, a lot of the quantitative measurements of protein expressions in this manuscript by the authors were using detection of the fluorescence intensity in the immunofluorescence staining assay. However, the estimation of MFI is a relative semi-quantitative test which is not accurate enough as the only verification method for protein expressions. In addition, the authors also did not specify the number of samples used, the area selected for each sample, and the measurement conditions. This part requires more precise measurement methods such as WB or qPCR to do further proof.

  Our team used all 3 types of cells together for isolating mRNA in the mRNA expression test because the mRNA isolation of each cell type is almost impossible in the 3D chip. We measured the expression of α-smooth muscle actin and the data described reversible patterns from each patient, so we selected data which have consistent pattern in every result. It may have different responses from each fibroblast according to patient severity or condition and fibroblast can have reversible character.  

2. Second, the author did not explain how to define the standard of thin and thick vessels, since the formation of a 3D tubular capillary-like network is an important experimental result of the author, they should explain in more detail. In addition, the author should examine the effects of SB4315422 on their experiments separately. Because the effects of TGF-β1+TGF-β1 inhibitor group is much higher than that of the control group, so the effects of SB4315422 alone

This is an important suggestion, We defined the standard of thin and thick vessels both Length and Diameter, Capillary-like network in case of shorter than 50 is thin vessel, and thick vessels are estimated in case of longer than 50. We added the definition to line149 on page 7 in revised manuscript.

The reason why we test SB4315422 was that we assumed that SB4315422 might be a positive control of potential anti-fibrotic candidate agents as a reference.

In most of the parameters, the effects of TGF-β1+TGF-β1 inhibitor group is almost similar or less than that of the control group, except vascular indicators of HUVEC, including total lengths and average of diameter.

Vascular indicators measured from 3 dimensional structure might be overestimated. Therefore we added the video clips of three groups in the supplemental, for better understanding the structural changes.

3. The authors described in conclusion that their 3D renal fibrosis-on-a-chip is better than the 2D culture experiments, but did not clearly explain what the characteristics of the 3D model are and how their model to simulate the situation of real kidney tissue? What the different and advantage is as comparing with co-culture together of 3 types of cells? Dose it belong 2D or 3D model in the 2.2 experiments on a chip? How is this different from the 2.3 experiments? just the addition of fibroblast in chip? This requires detailed explanation.

This is a good point, the result in Figure 1. (Result 2.1) is only (HK-2) 2D culture and the others are all 3D culture. Confocal images looks like 2D but 3D model in fact understandably except Figure 1. The results in Figure 2 (Result 2.2) are from HK-2 and HUVEC, 2 types of cells culture. The results from Figure 4 (Result 2.3) to Figure 6 express the addition of patient derived primary fibroblasts, 3 types of cells culture together at separate part in chip.

The advantage of this chip model is observation of each cells. We also mixed fibroblasts with primary epithelial cells and then plated in our 3D chip model in early days. It is almost impossible to distinguish the changes of each cell type like the pictures below, when we used the mixed cell even two types of cells in the chip. So we selected the results in manuscripts after several times experimental setup.

4. The authors need to explain which cell line they used for isolating mRNA in the mRNA expression test, HK2 or HUVECs? In addition, they also need to explain how the test determines the diameter of the thick and thin vessels in HUVECs. Is the increased number of cells taken into consideration?

This is a valid comment, we intended to express the techniques in general and revised, we used all 3 types of cells together for isolating mRNA in the mRNA expression test. The mRNA isolation of each cell type is almost impossible in the 3D chip. We determined the diameter of the thick and thin vessels in HUVECs, shorter than 50 is thin vessel, and thick vessels are estimated in case of longer than 50.

We edited in the definition to line 219 on page 12 in revised manuscript.

5. Although the author tested the HUVEC cell growth and cytokine growth factor expressions of TGF treatment in their 3D model, they did not clearly explain the exact effects of TGF-β1 on kidney fibrosis. By their results, is the fibroblast enhanced the TGF-β1 effects on renal fibrosis or TGF-β1 reduce the renal fibrosis in 3D model? The author seems to have only given an ambiguous description here, and make their conclusion about “3D renal fibrosis-on-a-chip could be used as potential in vitro model" become weak.

This suggestion is valid, we treated TGF-β1 in every experiment and anti-inflammatory action is one of the important roles of TGF-β1. We intended to describe the meaning related to inflammation together and fibroblast may act reversible by TGF-β1, because TGF-β1 have two different character.

The profibrotic and anti-inflammatory properties of TGF-β pose double-edged swords regarding the therapeutic application of TGF-β inhibition. Remarkable efforts have been made to obstruct TGF-β action to hinder the progression of renal fibrosis [3]. –line 38 on first page in manuscript.

TGF-β1 enhanced renal fibrosis and reduced the inflammation at the same time in our 3D model and these results are reasonable. In addition, we should admit that these fibrosis models might be created relatively in a short-term period, 24 hours. TGF-beta may behave two different characters, profibrotic and anti-inflammatory properties within an early fibrosis process, not in the chronic process. We added this limitation to the Discussion section, line 340 on page 16 in revised manuscript.

6. The immunofluorescent figures of the author showed are not clear enough, and higher resolution pictures should be provided for identification.

We apologize for this and I corrected in the manuscript file, and we revised the specific figure of resolution. The modified images may be transformed in Word and I will send the specific figures to the editorial office.

Summary of changes made to meet journal requirements.

Specifications of the final figures

Figure number

Figure type

Width (mm)

Height (mm)

Resolution (dpi)

File

format

Font type

Color mode

Figure 1

Combination art

190

90

500

.eps, .pdf, .tif

Arial

RGB

Figure 2

Combination art

190

138

500

.eps, .pdf, .tif

Arial

RGB

Figure 3

Combination art

190

120

500

.eps, .pdf, .tif

Arial

RGB

Figure 4

Combination art

190

112

500

.eps, .pdf, .tif

Arial

RGB

Figure 5

Line art

190

120

1200

.eps, .pdf, .tif

Arial

RGB

Figure 6

Line art

190

136

1200

.eps, .pdf, .tif

Arial

RGB

Reviewer 2 Report

This is a very interesting paper that seeks to develop an in vitro culture system that recreates some of the 3-dimensional architecture of the kidney to better test compounds that might block fibrosis in the kidney. The advantage of the platform is that it brings together three cell types that play a role in kidney fibrosis: epithelial cells, endothelial cells, and importantly interstitial fibroblasts.

  1. the authors perform 2D culture experiments to demonstrate the utility (sensitivity) of their platform but the work would be strengthened by including some time and dose response curves in both the 2D and 3D systems. The magnitude of the response of their 3D system (and its inhibition) might be improved by examining different doses. This would help in the eventual analyses of the ability to block fibrosis.
  2. The resolution of the IF images should be enhanced in Figures.
  3. I think that the research team should justify the design of the cell layers in their 3D chip. For example, they have interposed an endothelial cell layer between the fibroblasts and the epithelial cells but in the kidney, interstitial fibroblasts are not separated from epithelial cells by an endothelial cells layer. Endothelial mesenchymal transformation may occur, although this is not a focus of the paper while circulating fibrocytes may be recruited to the kidney and would initially be separated form the interstitial space by endothelial cells.
  4. The resolution of the images in Figure 4 is better that Figure 1 and 2 but again in the absence of dose response studies it is hard to know if the 3D cell culture system has been optimized.
  5. The work would be strengthened by better images showing the morphometric assessments of the endothelial cell. It was unclear from the image shown how these analyses were performed.
  6. The experimental groups in Figure 5 should be more clearly labelled to help with the interpretation.
  7. The work would be strengthened by extending the fibrosis read-out to measures of extracellular protein expression, for example, collagen I. This would complement the measures of α-smooth muscle actin. The focus of the work is on fibrosis so the analyses in Figure 6 that focus on inflammation does not directly address this goal. This work might best be removed from the paper.

Author Response

Response to Reviewer 2 Comments

Comments and Suggestions for Authors

This is a very interesting paper that seeks to develop an in vitro culture system that recreates some of the 3-dimensional architecture of the kidney to better test compounds that might block fibrosis in the kidney. The advantage of the platform is that it brings together three cell types that play a role in kidney fibrosis: epithelial cells, endothelial cells, and importantly interstitial fibroblasts.

We are very grateful for this precious comment and appreciate the positive responses.

  1. the authors perform 2D culture experiments to demonstrate the utility (sensitivity) of their platform but the work would be strengthened by including some time and dose response curves in both the 2D and 3D systems. The magnitude of the response of their 3D system (and its inhibition) might be improved by examining different doses. This would help in the eventual analyses of the ability to block fibrosis.                      This is an excellent point, we agree that it was only part of the content, there are a little different response between 2D and 3D, two models have some different conditions, so we evaluated additional time and concentration of TGF-β1 in 3D condition and the images are reflected in comment number 3.
  2. The resolution of the IF images should be enhanced in Figures.

We apologize for this and I corrected in the manuscript file, and we have revised the specific figure of resolution.

Summary of changes made to meet journal requirements.

Specifications of the final figures

Figure number

Figure type

Width (mm)

Height (mm)

Resolution (dpi)

File format

Font type

Color mode

Figure 1

Combination art

190

90

500

.eps, .pdf, .tif

Arial

RGB

Figure 2

Combination art

190

138

500

.eps, .pdf, .tif

Arial

RGB

Figure 3

Combination art

190

120

500

.eps, .pdf, .tif

Arial

RGB

Figure 4

Combination art

190

112

500

.eps, .pdf, .tif

Arial

RGB

Figure 5

Line art

190

120

1200

.eps, .pdf, .tif

Arial

RGB

Figure 6

Line art

190

136

1200

.eps, .pdf, .tif

Arial

RGB

3. I think that the research team should justify the design of the cell layers in their 3D chip. For example, they have interposed an endothelial cell layer between the fibroblasts and the epithelial cells but in the kidney, interstitial fibroblasts are not separated from epithelial cells by an endothelial cells layer. Endothelial mesenchymal transformation may occur, although this is not a focus of the paper while circulating fibrocytes may be recruited to the kidney and would initially be separated form the interstitial space by endothelial cells.    

This is an important suggestion, we mixed fibroblasts with primary epithelial cells and then plated in our 3D chip model in early days. We selected the results in manuscripts after several times experimental setup. It is almost impossible to distinguish each cells changes when we used the mixed cell.  

4. The resolution of the images in Figure 4 is better that Figure 1 and 2 but again in the absence of dose response studies it is hard to know if the 3D cell culture system has been optimized.   

We have revised and described the results in response to comment number 1 to 3. The modified images may be transformed in Word and I will send the specific figures to the editorial office.

5. The work would be strengthened by better images showing the morphometric assessments of the endothelial cell. It was unclear from the image shown how these analyses were performed.We defined the standard of thin and thick vessels both Length and Diameter, Capillary-like network in case of shorter than 50 is thin vessel, and thick vessels are estimated in case of longer than 50.      

We added to line149 on page 7 - Legend of Figure 2 and line 219 on page 12 - Legend of Figure 4 in revised manuscript.

6. The experimental groups in Figure 5 should be more clearly labelled to help with the interpretation.  

Thank you for your fine advice, we edited in the label into the Figure 5.

7. The work would be strengthened by extending the fibrosis read-out to measures of extracellular protein expression, for example, collagen I. This would complement the measures of α-smooth muscle actin. The focus of the work is on fibrosis so the analyses in Figure 6 that focus on inflammation does not directly address this goal. This work might best be removed from the paper.

This suggestion is valid, we treated TGF-β in every experiment and anti-inflammatory action is one of the important roles of TGF-β, we intended to describe the meaning related to inflammation together and fibroblast may act reversible by TGF-β1, because TGF-β1 have two different character.

The profibrotic and anti-inflammatory properties of TGF-β are double-edged swords regarding the therapeutic application of TGF-β inhibition. Remarkable efforts have been made to obstruct TGF-β action to hinder the progression of renal fibrosis [3]. –line 38 on first page in manuscript.

Fibrosis is the end result of chronic inflammatory reactions induced by a variety of stimuli including persistent infections, autoimmune reactions, allergic responses, chemical insults, radiation, and tissue injury.  https://www.ncbi.nlm.nih.gov › articles › PMC2693329 

 TGF-β1 enhanced renal fibrosis and reduced the inflammation at the same time in our 3D model and these results are reasonable but we realized that the therapeutic application of TGF-β inhibition is double edged swords.  

In addition, we admit that these fibrosis models might be created relatively in a short-term period, 24 hours. TGF-beta may behave two different characters, pro-fibrotic and anti-inflammatory properties within an early fibrosis process, not in the chronic process. We added this limitation in the Discussion section, to line 342 on page 16 in revised manuscript.

 We intended to express the generalized results and our team used all 3 types of cells together for isolating mRNA in the mRNA expression test and measured α-smooth muscle actin and the data described reversible patterns from each patient, so we selected data which have consistent pattern in every result. It may have different responses from each fibroblast according to patient severity or condition and fibroblast also can have reversible character.

Round 2

Reviewer 1 Report

All the comments of me have been addressed well.

Reviewer 2 Report

Revisions are satidfcatory.